

# Concordance of chest x-ray with chest CT by body mass index

Thomas F. Heston[1,2] and John Y. Jiang[2,3]

[1] Primary Care, Mann-Grandstaff Veterans Administration Medical Center, Spokane, Washington, United States
[2] Medical Education and Clinical Sciences, Washington State University, Spokane, Washington, United States
[3] Diagnostic Imaging, Mann-Grandstaff Veterans Administration Medical Center, Spokane, Washington, United States

Corresponding author
Thomas F. Heston,
tomhestonmd@gmail.com

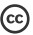

## ABSTRACT

**Introduction:** Patients with suspected thoracic pathology frequently get imaging with conventional radiography or chest x-rays (CXR) and computed tomography (CT). CXR include one or two planar views, compared to the three-dimensional images generated by chest CT. CXR imaging has the advantage of lower costs and lower radiation exposure at the expense of lower diagnostic accuracy, especially in patients with large body habitus.

**Objectives:** To determine whether CXR imaging could achieve acceptable diagnostic accuracy in patients with a low body mass index (BMI).

**Methods:** This retrospective study evaluated 50 patients with age of $63 \pm 12$ years old, 92% male, BMI $31.7 \pm 7.9$, presenting with acute, nontraumatic cardiopulmonary complaints who underwent CXR followed by CT within 1 day. Diagnostic accuracy was determined by comparing scan interpretation with the final clinical diagnosis of the referring clinician.

**Results:** CT results were significantly correlated with CXR results ($r = 0.284$, $p = 0.046$). Correcting for BMI did not improve this correlation ($r = 0.285$, $p = 0.047$). Correcting for BMI and age also did not improve the correlation ($r = 0.283$, $p = 0.052$), nor did correcting for BMI, age, and sex ($r = 0.270$, $p = 0.067$). Correcting for height alone slightly improved the correlation ($r = 0.290$, $p = 0.043$), as did correcting for weight alone ($r = 0.288$, $p = 0.045$). CT accuracy was 92% (SE = 0.039) *vs.* 60% for CXR (SE = 0.070, $p < 0.01$).

**Conclusion:** Accounting for patient body habitus as determined by either BMI, height, or weight did not improve the correlation between CXR accuracy and chest CT accuracy. CXR is significantly less accurate than CT even in patients with a low BMI.

## INTRODUCTION

The sensitivity and specificity of chest CT is superior to chest x-ray (CXR) across nearly all diagnostic categories but is associated with patient exposure to about 50 times more ionizing radiation. A CXR exposes patients to about 0.1 mSv compared to about 6.1 mSv for chest CT (*Health Physics Society, 2021*). To minimize iatrogenic harm, save time, and

minimize costs, CXR is often the imaging modality of choice for emergency room and urgent care patients being evaluated for respiratory complaints. In some cases, this may not be the best way to proceed. One study of 3,423 emergency department patients undergoing both CXR and chest CT found that when using CT as the diagnostic standard for pulmonary opacities, CXR had a sensitivity of only 44% and specificity of 93% (*Self et al., 2013*). In 42 children hospitalized with complicated pneumonia, CXR had a accuracy of just 42% compared to CT in the assessment of complications (*Tan Kendrick et al., 2002*). Another study looked at the sensitivity and specificity of CXR compared with CT in the diagnosis of COVID-19 using reverse transcriptase PCR as the gold standard. In this study of 1,198 patients, CXR had an accuracy of 57% whereas CT had an accuracy of 79%, the researchers considered the agreement between CT and CXR to be poor with a Cohen's kappa of 0.406 (*Borakati et al., 2020*). Although time considerations in the emergency department may favor the routine use of CXR before CT, even in routine outpatient settings the American College of Radiology Appropriateness Criteria recommends that chest CT be utilized only as a follow-up study after a patient has had an initial CXR (*Jokerst et al., 2018*; *American College of Radiology, 2021*). Although significant differences in diagnostic accuracy exist, the recommendation remains to have the CXR act as a gatekeeper for a chest CT.

Given the low sensitivity of CXR in emergency department patients, it is hypothesized that an elevated body mass index may lower the threshold for obtaining a chest CT in addition to CXR. If so, there may be a cutoff point that would strongly indicate the need for CT imaging in addition to plain radiography.

## MATERIALS AND METHODS

A retrospective review of existing medical records was performed. Patients with a CXR and chest CT within 24 h were included in the study. Scan interpretation was compared with the final clinical diagnosis. Using the final clinical diagnosis as the gold standard, scan results were coded as positive (showing disease) or negative (normal). Results were further coded as true positive (TP), true negative (TN), false positive (FP), or false negative (FN). Patient weight, height, age, and sex were recorded.

The relationship between CXR results and CT results were evaluated with the bivariate Spearman rank correlation coefficient (r). Partial correlation was utilized to control separately for height, weight, and BMI. Partial correlation was also utilized to control for both BMI and age together. Phi kappa was utilized to gauge the association between CXR and CT findings when looking at TP, FP, TN, and FN categories.

For sensitivity and specificity analyses, disease was categorized as (a) patients with a vascular diagnosis (congestive heart failure); (b) patients with a respiratory diagnosis (pneumonia, chronic obstructive pulmonary disease, pulmonary embolism, or bronchitis); or patients with the combined endpoint of either a respiratory or vascular diagnosis.

Analyses were performed using IBM SPSS Statistics Version 28 (SPSS, Inc., Chicago, IL, USA). This study was approved with individual consent waived by the Veterans Administration Puget Sound IRB (protocol #1608973).

**Table 1 Baseline characteristics.**

|  | Mean (std) | Minimum | Maximum |
|---|---|---|---|
| Age | 63.8 (12.0) | 35 | 94 |
| Height (cm) | 176.2 (8.3) | 157.5 | 193.0 |
| Weight (kg) | 98.2 (24.3) | 63.7 | 169.2 |
| BMI (kg/m$^2$) | 31.7 (7.8) | 20.2 | 58.4 |
| Sex | 92% male |  |  |
| Total subjects | 50 |  |  |

**Note:**
Table entries are given as the mean (standard deviation).

## RESULTS

There were 50 cases included in the study. Age was 63.8 years with a rage of 35 to 94, a standard deviation (std) of 12.0, and a 95% CI [60.5–67.2]. Most patients were male (46/50 = 92%). BMI was 31.7 kg/m$^2$ with a range of 20.2 to 58.4, a std 7.8, and a 95% CI [29.5–33.8] (Table 1).

Patient body habitus as measured by BMI was in the healthy weight range (BMI of 18.5 to 24.9) in eight subjects (16%); overweight (BMI 25.0 to 29.9) in 12 subjects (24%), and obese (BMI of 30.0 or higher) in 30 patients (60%). A positive CXR was followed by a negative CT in 28% of cases (7/25). A negative CXR was followed by a positive CT in 44% of cases (11/25).

CXR findings were positive for disease in 25/50 patients and CT findings were positive for disease in 29/50 patients. These findings were significantly correlated (r = 0.284, $p = 0.045$, degree of freedom (df) = 48). Controlling separately for BMI, weight, or height only minimally affected this correlation (r = 0.285, $p = 0.047$, df = 47; r = 0.288, $p = 0.045$, df = 47; r = 0.290, $p = 0.043$, df = 47). Controlling for BMI and age together also did not change the correlation (r = 0.283, $p = 0.052$, df = 46).

CXR findings were correct (true positive or true negative) in 30 patients and incorrect (false positive or false negative) in 20 patients. CT findings were correct in 46 patients and incorrect in four patients. When categorized as correct or incorrect, CXR findings were not significantly associated with CT findings (r = −0.090, $p = 0.53$, df = 48). Controlling separately for BMI, weight, or height did not improve the correlation (r = −0.095, $p = 0.516$, df = 47; r = −0.092, $p = 0.528$, df = 47; r = −0.088, $p = 0.548$, df = 47). Controlling for BMI and age together also did not change the correlation (r = −0.098, $p = 0.506$, df = 46).

When imaging results were categorized as true positive, true negative, false positive, or false negative, the CXR findings agreed with the CT findings in 27 of 50 patients (Table 2). This association was significant (kappa = 0.932, $p < 0.001$, df = 1). When patients were categorized as obese (BMI > 30) or not obese (BMI < 30), the association was slightly more significant in the obese ($n = 30$, kappa = 0.931, $p = 0.002$) compared to the non-obese ($n = 20$, kappa = 0.947, $p = 0.006$).

Out of the 50 patients, 12 had a vascular diagnosis and 21 had a respiratory diagnosis. The other 17 diagnoses included musculoskeletal pain (*American College of Radiology,*

**Table 2 CXR vs. chest CT results stratified by diagnostic category.**

| | | CT findings | | | | |
|---|---|---|---|---|---|---|
| | | FN | FP | TN | TP | Total |
| CXR findings | FN | 1 | 0 | 0 | 9 | 10 |
| | FP | 0 | 0 | 7 | 3 | 10 |
| | TN | 0 | 2 | 13 | 0 | 15 |
| | TP | 0 | 1 | 0 | 14 | 15 |
| | Total | 1 | 3 | 20 | 26 | 50 |

Note:
FN, false negative; FP, false positive; TN, true negative; TP, true positive.

**Table 3 Diagnostic performance of CXR and chest CT for respiratory diseases, vascular diseases, and the combined endpoint of respiratory or vascular disease.**

| | Sensitivity | Specificity | Accuracy | Prevalence |
|---|---|---|---|---|
| | Respiratory diseases | | | |
| CXR | 0.57 [0.36–0.78] | 0.55 [0.37–0.73] | 0.56 [0.42–0.70] | 0.42 |
| CT | 0.76 [0.58–0.94] | 0.55 [0.37–0.73] | 0.64 [0.51–0.77] | – |
| | Vascular diseases | | | |
| CXR | 0.33 [0.07–0.60] | 0.45 [0.29–0.61] | 0.42 [0.28–0.56] | 0.24 |
| CT | 0.42 [0.14–0.70] | 0.37 [0.21–0.52] | 0.38 [0.25–0.51] | – |
| | Respiratory or vascular diseases | | | |
| CXR | 0.48 [0.31–0.66] | 0.47 [0.23–0.71] | 0.48 [0.34–0.62] | 0.66 |
| CT | 0.64 [0.47–0.80] | 0.53 [0.29–0.77] | 0.60 [0.46–0.74] | – |

Note:
Table entries are given as the value (95% confidence interval).

**Table 4 Predictive values and likelihood ratios for CXR and chest CT for respiratory disease, vascular disease, or the combined endpoint of respiratory or vascular disease.**

| | PPV | NPV | LR+ | LR− |
|---|---|---|---|---|
| | Respiratory diseases | | | |
| CXR | 0.48 [0.28–0.68] | 0.64 [0.45–0.83] | 1.275 [0.74–2.20] | 0.777 [0.43–1.41] |
| CT | 0.55 [0.37–0.73] | 0.76 [0.58–0.94] | 1.700 [1.06–2.72] | 0.432 [0.19–0.99] |
| | Vascular diseases | | | |
| CXR | 0.16 [0.02–0.30] | 0.68 [0.50–0.86] | 0.603 [0.26–1.41] | 1.490 [0.87–2.54] |
| CT | 0.17 [0.03–0.31] | 0.67 [0.47–0.87] | 0.66 [0.32–1.34] | 1.583 [0.84–2.99] |
| | Respiratory or vascular diseases | | | |
| CXR | 0.64 [0.45–0.83] | 0.32 [0.14–0.50] | 0.916 [0.52–1.62] | 1.095 [0.60–2.00] |
| CT | 0.72 [0.56–0.87] | 0.43 [0.22–0.64] | 1.352 [0.77–2.38] | 0.687 [0.36–1.30] |

Note:
Table entries are given as the value (95% confidence interval).

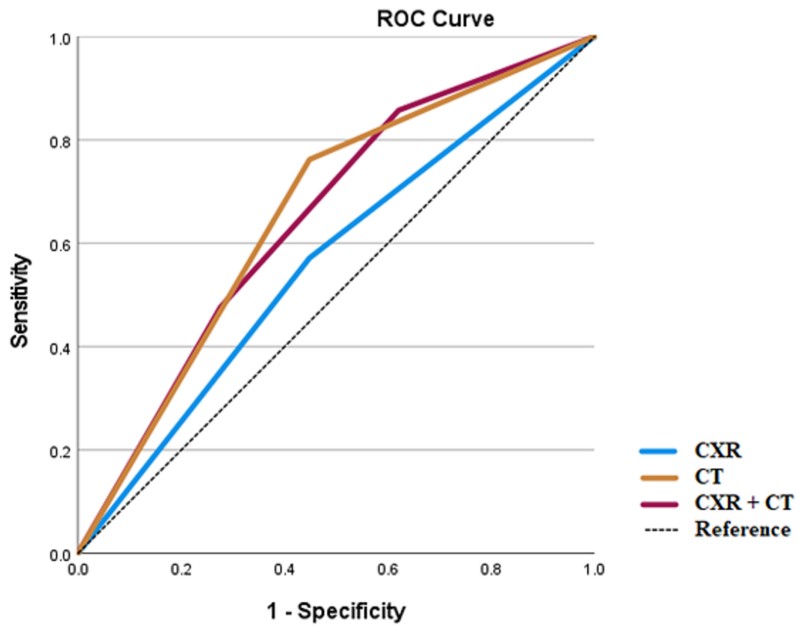

**Figure 1 ROC analysis for respiratory diseases.**

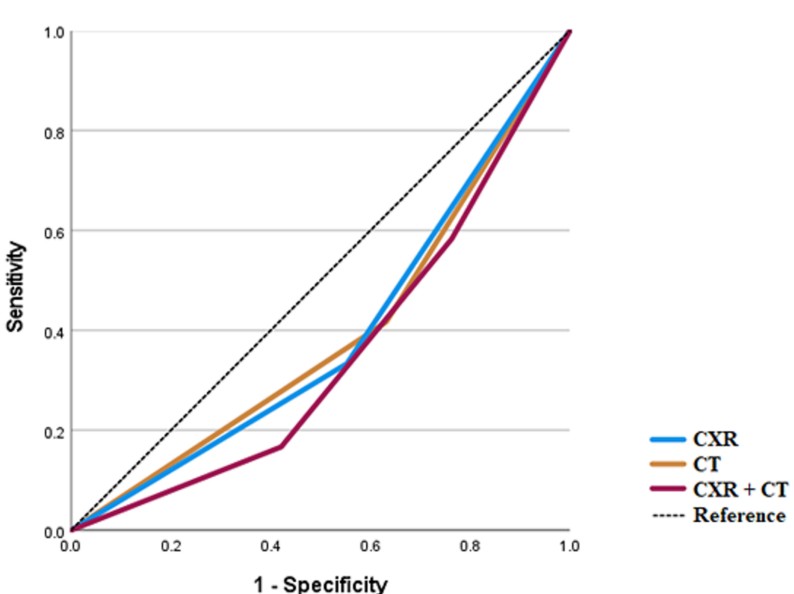

**Figure 2 ROC analysis for vascular diseases.**

*2021*), cancer (*Jokerst et al., 2018*), non-respiratory infection (*Borakati et al., 2020*), polypharmacy (*Health Physics Society, 2021*), and normal (*Health Physics Society, 2021*). An analysis was performed looking at the diagnostic performance of CXR and chest CT for the endpoints of respiratory diseases, vascular diseases, or the combined endpoint of vascular or respiratory diseases (Table 3). Predictive values and likelihood ratios were also calculated (Table 4). Overall, in our sample of 50 patients with a prevalence of disease ranging from 0.24 to 0.66, the diagnostic performance of CXR compared to chest CT were not significantly different.

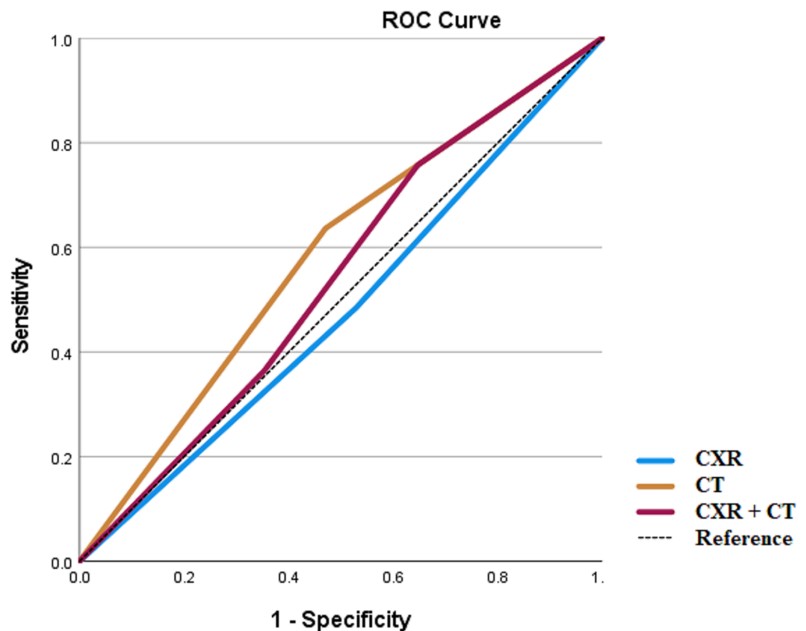

**Figure 3** **ROC analysis for respiratory or vascular diseases.**

CXR and chest CT results were utilized for an ROC curve analysis for the endpoints of respiratory disease, vascular disease, or the combined endpoint of vascular or respiratory disease. In addition, for this analysis a third variable was created, CXRCT, which took on three values: 0 if both CXR and CT results were negative; one if either the CXR or CT results were positive; and two if both the CXR and CT results were positive.

Both CXR and chest CT performed the best for the diagnosis of respiratory disease (Fig. 1). For respiratory disease, the area under the curve (AUC) for CXR was 0.562 (standard error (SE) 0.083, $p = 0.461$), for CT 0.657 (SE 0.078, $p = 0.060$) and for CXRCT 0.648 (SE 0.078, $p = 0.077$).

Both CXR and chest CT underperformed for the diagnosis of vascular disease (Fig. 2). For vascular disease, the AUC for CXR was 0.390 (SE 0.093, $p = 0.256$), for CT 0.393 (SE 0.095, $p = 0.266$), and for CXRCT 0.351 (SE 0.088, $p = 0.122$).

For the combined endpoint of respiratory or vascular disease, the AUC again was greatest for CT (Fig. 3). The AUC for CXR was 0.478 (SE 0.087, $p = 0.798$), for CT 0.583 (SE 0.086, $p = 0.341$), and for CXRCT 0.539 (0.089, $p = 0.652$).

In all cases, the ROC analysis showed that the AUC for chest CT was greater than for CXR or for CXRCT.

## DISCUSSION

This study found that a patient's BMI did not affect the accuracy of CXR findings. BMI also did not affect the accuracy of chest CT findings. Finally, BMI did not affect the concordance between CXR and chest CT findings. Clinicians should not let a patient's BMI affect whether the patient undergoes a CXR, a chest CT, or both. A high BMI did not make imaging less accurate, and a low BMI did not make imaging more accurate.

Previous studies have shown a correlation between body mass index and image quality. One study looking at cardiac computed tomography found that the signal to noise ratio was higher in patients with a BMI of under 30 kg/m$^2$ compared to over 30 mg/m$^2$, however, the diagnostic accuracy of the CT was good regardless of BMI (*Latif et al., 2016*). This study demonstrated that while technical metrics of image quality were affected by BMI, the diagnostic accuracy was not.

The effects of age and sex upon the accuracy of CXR scores in the diagnosis of SARS-CoV-2 infection have shown conflicting results. In one retrospective study of Mexican-mestizo patients, there was no difference in the total CXR score between males and females grouped by age (*Albrandt-Salmeron, Espejo-Fonseca & Roldan-Valadez, 2021*). In a different study of Italian patients hospitalized with SARS-CoV-2 infection, the CXR score was positively associated with age in both males and females (*Borghesi et al., 2020*). In our veteran population, a breakdown of results by sex was not possible given that only four out of 50 patients were female. However, we found that age did not affect the correlation between CXR and CT.

One possibility that would explain the lack of effect of BMI on scan accuracy is that the adjustment of radiation exposure utilized by technologists and equipment is just right, increasing the radiation exposure by the amount necessary to maintain image quality. However, it appears likely that CT automatic exposure controls based upon BMI over-expose patients to radiation. These automated adjustments made by the CT software may be able to be significantly improved by basing adjustments on minimum radiation dosages (*Cho et al., 2018*), by adjustments based on patient girth at the location of imaging (*Glanc et al., 2012*), or by improved reconstruction techniques (*Sulieman et al., 2021*).

Given the significant difference found in diagnostic accuracy between CXR and chest CT, it may be that CT imaging may replace conventional radiography for many clinical indications. For example, one study looking at ultra-low dose CT found that for the evaluation of pulmonary emphysema, the diagnostic quality was equal to regular dose CT in spite of a 95% reduction in radiation exposure, from 2.33 mSv down to 0.12 mSv (*O'Brien et al., 2019*). Another study looking at cervical spine imaging found that conventional radiography could be replaced with a nearly dose-neutral CT scan (*Deak et al., 2022*).

This study confirms the superior accuracy of CT imaging, and when a chest CT is ordered, that the CXR adds little if any additional diagnostic value. One primary value of CXR is the speed of acquisition, especially in critically ill patients. This rapid overview of the chest can tailor subsequent imaging and attention. On the other hand, patients that are sick enough to require admission to the hospital may benefit from early ordering of chest CT imaging.

Overall, our study found low diagnostic performance for both CXR and CT imaging. It is often thought that diagnostic performance of a test is independent of disease prevalence in terms of sensitivity and specificity, while predictive values are highly dependent upon disease prevalence. Empirical studies, however, have frequently shown that sensitivity, specificity, and accuracy of a test are also highly dependent upon disease prevalence (*Brenner & Gefeller, 1997*). As the focus of this study was to determine the

correlation between CXR and CT for all-comers, the prevalence of any specific disease was low, ranging from 0.24 for vascular disease to 0.42 for respiratory disease. This may in large part be responsible for the overall low diagnostic performance of both CXR and CT scanning found in our study. This finding raises the possibility that the best way to reduce patient radiation exposure is by more stringent thresholds to order CXR or CT imaging. Using a strategy of routine imaging to simply rule out disease is appealing to clinicians, but likely leads to over testing resulting in poor diagnostic performance and unnecessary radiation exposure to patients.

Study limitations include the relatively small sample size and the difficulty of categorizing scan results into true positive, true negative, false positive, or false negative. While large sample sizes can pick up small differences in patient populations, this sensitivity for differences often becomes clinically meaningless (*Lantz, 2013*). This study looked at effect sizes to account for sample size, because effect sizes are not dependent upon sample size. The effect sizes observed confirm our conclusions that: (a) CT is much more accurate than CXR, (b) that the concordance between CT and CXR was moderate, and (c) the relationship between BMI and scan accuracy (both CT and CXR) is weak.

Categorizing scan findings as true or false is difficult because of intimate relationship between imaging findings and ultimate clinical diagnosis and management decisions. This study is unique in that it is not just a database review of scan findings and final diagnosis codes. Rather, each patient was individually reviewed, looking closely at scan findings and clinical course. By reviewing the clinical course, it is possible to determine whether initial treatment decisions based on imaging resulted in an expected clinical outcome or not. Nevertheless, categorizing scans as true or false remains challenging not only because scan results strongly bias clinical management, but also because often patients get better or worse regardless of the treatment rendered. Also, frequently clinicians will simultaneously treat multiple conditions. For example, a patient who is in heart failure might be treated with both diuretics and antibiotics based upon a CT showing suspected pneumonia but clinical findings of heart failure. In such a case, it is nearly impossible to determine whether the conditions co-existed, or if the patient had only one of the two conditions. Nevertheless, one of the strengths of this study is that by individual review of the patient's clinical course, both the treating clinician's decision-making and ultimate patient outcome can be fairly evaluated. By close chart review, not just a database review of ICD-10 codes, greater accuracy in categorizing scan results is possible.

Our study is also limited by the lack of reporting for radiation levels utilized for CT imaging. However, when controlling for BMI, weight, or height, the largest gain in correlation between CXR and CT imaging was found when controlling for height alone, not for BMI. Reduction of radiation exposure by adjusting CT scanner tube current based upon BMI is one method used in chest CT scanning (*Cho et al., 2018*; *Brat et al., 2019*; *Manowitz et al., 2012*). Although our evidence is weak regarding this issue, it does raise the possibility that adjusting CT radiation levels by BMI may not be the optimal strategy and that greater attention to body habitus and fat distribution will better enable radiation dose reductions without negatively affecting image quality.

This study is also limited by the demographic makeup of the study population. The patients were exclusively US veterans, which were almost entirely male. This made any analysis based upon sex highly fragile with very low accuracy.

Overall, this study found that CXR and chest CT scan accuracy under current imaging protocols is not significantly affected by a patient's BMI.

## ACKNOWLEDGEMENTS

The authors would like to acknowledge the Research & Development Committee at the Mann-Grandstaff Veterans Administration Medical Center for administrative and technical guidance.

### Funding
The authors received no funding for this work.

### Competing Interests
The authors declare that they have no competing interests.

### Author Contributions
- Thomas F. Heston conceived and designed the experiments, performed the experiments, analyzed the data, prepared figures and/or tables, authored or reviewed drafts of the article, and approved the final draft.
- John Y. Jiang analyzed the data, authored or reviewed drafts of the article, and approved the final draft.

### Human Ethics
The following information was supplied relating to ethical approvals (*i.e.*, approving body and any reference numbers):

This study was approved the VA Puget Sound Institutional Review Board (protocol #1608973).

### Data Availability
The data available on Figshare: Heston, Thomas; Jiang, John (2023): Chest XR *vs.* chest CT concordance. figshare. Dataset. https://doi.org/10.6084/m9.figshare.22082714.v1.

### Supplemental Information
Supplemental information for this article can be found online at http://dx.doi.org/10.7717/peerj.15090#supplemental-information.

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
