# Peer review of "Concordance of chest x-ray with chest CT by body mass index"

_PeerJ, doi:10.7717/peerj.15090_

## Round 0.1 · original submission · Major Revisions

Dear Authors,

The reviewers and I have completed our evaluation of your manuscript and recommend a major revision before re-submission.

Please review the comments and resubmit your revised manuscript.

·

Basic reporting

The authors wrote a fascinating manuscript on a needed topic. This kind of manuscript dealing with problems of day-to-day practice should receive attention from the global community.

Experimental design

This reviewer believes that the research question should be redefined to calculate the correlation coefficient between chest X-rays and Chest CT; after controlling the effect of BMI and other confounding variables like age.
The sample size of 50 is enough to run a Spearman correlation. Authors could code the four diagnosis statuses as 1 to 4 for both methods and then run a PARTIAL correlation analysis. First, in moment zero (without controlling the effect of controlling variables) and then a moment 1, the proper partial correlation by controlling the impact of body mass index and in a second analysis maintaining together BMI and age. The key to the study is to compare the correlation coefficient before and after controlling this effect. It is also essential to mention in the introduction and methods that recent papers have correlated chest X-rays with severity and comorbidities with significant findings using this method. Considered to include the missing reference:

Correlation between Chest X-Ray Severity in COVID-19 and Age in Mexican-Mestizo Patients: An Observational Cross-Sectional Study. Biomed Res Int. 2021 Apr 29;2021:5571144. Doi: 10.1155/2021/5571144. PMID: 33997012; PMCID: PMC8090453.

Another option would be to run complete diagnostic test performances of chest x-rays against the gold standard you chose. In 2nd step, the same analysis for the chest CT against the gold standard, the author can compare the AUROC and other values (accuracy, sensitivity, specificity, likelihood ratios, and predictive values. A complete diagnostic performance of chest x-rays is missing in current literature, and I believe it will be of value to clinicians in emergency departments.

It is essential to mention that both correlation results can have 95% confidence intervals and each diagnostic test performance. An example of how to present the tables can be found in the below reference:

Diagnostic performance of CT densities in selected grey- and white-matter regions for the clinical diagnosis of brain death: A retrospective study in a tertiary-level general hospital. Eur J Radiol. 2018 Nov;108:66-77. doi: 10.1016/j.ejrad.2018.09.023. Epub 2018 Sep 19. PMID: 30396673.

A partial correlation analysis is not standard in the literature. Still, for this manuscript, this reviewer believes it can be an analysis that brings more value and would strengthen the authors' findings. An example of how to report tables or figures with partial correlation can be found below:

Partial correlation analyses of global diffusion tensor imaging-derived metrics in glioblastoma multiforme: Pilot study. World J Radiol. 2015 Nov 28;7(11):405-14. doi: 10.4329/wjr.v7.i11.405. PMID: 26644826; PMCID: PMC4663379.

Validity of the findings

Conclusions were linked to the original research question. New findings are expected after the authors perform the further proposed analyses.

Additional comments

This reviewer wants to congratulate the author for probing whether it is possible to run clinical research with day-to-day examinations in an emergency department.

Reviewer 2 ·

Basic reporting

This study tried to determine the concordance of chest X-Ray with Chest CT by BMI. The results show that a) CT is much more accurate than CXR, b) that the concordance between CT and CXR was moderate, and c) the relationship between BMI and scan accuracy (both CT and CXR) is weak. The paper is well presented from the research meaning and the overall frame, but still has some deficiencies that should be corrected.

Experimental design

This paper evaluated 50 patients with BMI 31.7 ± 7.9, and tried to determine whether CXR imaging could achieve acceptable diagnostic accuracy in patients with a low BMI. However, BMI in the range of 31.7 ± 7.9 is not a range of low BMI. According to https://www.cdc.gov/healthyweight/assessing/index.html, BMI in the range of 31.7 ± 7.9 falls within the overweight range (the high BMI group). The description of the enrolled subjects should be carefully checked.

Validity of the findings

The author claimed that the relationship between BMI and scan accuracy (both CT and CXR) is weak. Meanwhile, the author pointed out that the automated adjustments of radiation intensity by scanning equipment overcomes any effect of BMI. But the effects of radiation intensity on scan accuracy were not discussed in the paper. So, a statistical analysis for the relationship between radiation intensity and scan accuracy should be take into consideration.

---

## Round 0.2 · accepted · Accept

Your manuscript has been accepted for publication. Congratulations!

·

Basic reporting

no comment

Experimental design

no comment

Validity of the findings

no comment

Additional comments

The authors have adequately addressed the remarks made by this reviewer. The recommendation of this reviewer is acceptance of the manuscript.

Reviewer 2 ·

Basic reporting

No comment

Experimental design

No comment

Validity of the findings

No comment

Additional comments

The authors successfully addressed the all of concerns raised by me.